# Global Distribution of Founder Variants Associated with Non-Syndromic Hearing Impairment

**DOI:** 10.3390/genes14020399

**Published:** 2023-02-03

**Authors:** Elvis Twumasi Aboagye, Samuel Mawuli Adadey, Edmond Wonkam-Tingang, Lucas Amenga-Etego, Gordon A. Awandare, Ambroise Wonkam

**Affiliations:** 1West African Centre for Cell Biology of Infectious Pathogens (WACCBIP), University of Ghana, Accra LG Box 54, Ghana; 2Division of Human Genetics, Faculty of Health Sciences, University of Cape Town, Cape Town 7925, South Africa; 3McKusick-Nathans Institute and Department of Genetic Medicine, John Hopkins University School of Medicine, Baltimore, MD 21205, USA

**Keywords:** hearing impairment, non-syndromic, genetics, founder variant, gab junction beta 2 *(GJB2*), global populations

## Abstract

The genetic etiology of non-syndromic hearing impairment (NSHI) is highly heterogeneous with over 124 distinct genes identified. The wide spectrum of implicated genes has challenged the implementation of molecular diagnosis with equal clinical validity in all settings. Differential frequencies of allelic variants in the most common NSHI causal gene, gap junction beta 2 (*GJB2*), has been described as stemming from the segregation of a founder variant and/or spontaneous germline variant hot spots. We aimed to systematically review the global distribution and provenance of founder variants associated with NSHI. The study protocol was registered on PROSPERO, the International Prospective Register of Systematic Reviews, with the registration number “CRD42020198573”. Data from 52 reports, involving 27,959 study participants from 24 countries, reporting 56 founder pathogenic or likely pathogenic (P/LP) variants in 14 genes (*GJB2*, *GJB6*, *GSDME*, *TMC1*, *TMIE*, *TMPRSS3*, *KCNQ4*, *PJVK*, *OTOF*, *EYA4*, *MYO15A*, *PDZD7*, *CLDN14*, and *CDH23*), were reviewed. Varied number short tandem repeats (STRs) and single nucleotide polymorphisms (SNPs) were used for haplotype analysis to identify the shared ancestral informative markers in a linkage disequilibrium and variants’ origins, age estimates, and common ancestry computations in the reviewed reports. Asia recorded the highest number of NSHI founder variants (85.7%; 48/56), with variants in all 14 genes, followed by Europe (16.1%; 9/56). *GJB2* had the highest number of ethnic-specific P/LP founder variants. This review reports on the global distribution of NSHI founder variants and relates their evolution to population migration history, bottleneck events, and demographic changes in populations linked with the early evolution of deleterious founder alleles. International migration and regional and cultural intermarriage, coupled to rapid population growth, may have contributed to re-shaping the genetic architecture and structural dynamics of populations segregating these pathogenic founder variants. We have highlighted and showed the paucity of data on hearing impairment (HI) variants in Africa, establishing unexplored opportunities in genetic traits.

## 1. Introduction

Despite the multifactorial etiology of non-syndromic hearing impairment (NSHI), genetic contribution accounts for from 50 to 60% of congenital hearing impairment (HI), with high genetic and allelic heterogeneity [1]. Of the array of known NSHI-associated genes, pathogenic variants in the gene encoding the gap junction beta 2 protein (GJB2) remain the most recurrent cause of NSHI across populations [2]. HI genetic screening and/or testing in at-risk newborns provides evaluative information for accurate diagnosis, for early detection, management, and anticipatory guidance [3]. However, the wide genetic heterogeneity and differential allelic frequencies in NSHI causal markers across populations continue to challenge the implementation of a universal molecular diagnostic tool with equal clinical validity in all settings. 

Moreover, in spite of the extensive and systematic investigation of implicated NSHI candidate genes in the past two decades, the global equity in regional HI genetic research investment can be improved, as the quality and quantity of studies from some regions of the world, including Africa, are far behind [4]. Among the reported HI-implicated genes, variations in the gap junction beta 2 gene (*GJB2*) (MIM*121011), solute carrier family 26 member 4 (*SLC26A4*) (MIM*605646), and mitochondrial 12S rRNA (*MT-RNR1*) (MIM*561000) are the most associated genes across populations [5]. However, whereas *GJB2*: c.35delG-p.Gly12ValfsTer2 is the predominant NSHI causal variant among Europeans, North Africans, Brazilians, and Americans [6,7,8], the *GJB2*: c.235delC-p.Leu79CysfsTer3, c.845G>A-p.Met163Val, c.109G>A-p.Val37Ile, c.167delT-p.Lys56Arg, c.71G>A-p.Trp24X [9], and c.427 C>T-p.Arg143Trp [10,11,12] are prevalent in Asia, Ashkenazi Jews, India, and Ghana, respectively. Similarly, the *GJB2*: c.-23+1G>A is the most common NSHI variant reported in Southwest [13] and South Asians [14]. Equally, variations in *SLC26A4* are regional and population-specific; while *SLC26A4*: c.1246A>C-p.Thr416Pro and c.1001G>A-p.Gly334Glu are mostly reported in Europe [13,15], the c.2168A>G-p.His723Arg is common in Japan [16] and Korea [17], with c.919-2A>G reported in Taiwan [18] and Han Chinese populations [19]. 

The observed differential incidence of deleterious allele segregation with NSHI across populations is largely shaped by genetic architecture and population structural dynamics, including the rate of mutations, immigration/migration, admixture, and unknown environmental pressure that could have led to the likely selection of some pathogenic alleles, described as founder mutations. However, to date, the evolutionary forces driving the persistent segregation of reported deleterious NSHI founder variants across populations remains to be fully elucidated. It is imperative to investigate and generate data to better understand the probable evolutionary constraints around the genomic regions of these founder loci. This will further highlight the effective contribution of population size on founder phenomena. We asked the fundamental question as to whether migration, demographic history, and bottleneck events in the history of ethnic groups with higher frequencies of these founder alleles, may have led to the evolution and subsequent segregation of NSHI markers. 

In this paper, we have reviewed the distributions and provided a global profile of the reported NSHI founder variants’ prevalence, origins, and estimated ages across populations. The study also demonstrated and defined the ancestral evolution and ethnic dynamics of NSHI founder variants while contributing to refining the regional molecular diagnostic tools, as well as setting our future research agenda.

## 2. Materials and Methods

We conducted this systematic review of NSHI-associated founder variants following the guidelines of the Preferred Reporting Items for Systematic Reviews and Meta-Analyses (PRISMA) [20]. In addition, the study protocol was registered on PROSPERO, the International Prospective Register of Systematic Reviews with the registration number; “CRD42020198573”.

### 2.1. Search Strategy

The authors performed a literature search on electronic databases (PubMed, Scopus, Google Scholar, Directory of Open Access Journal (DOAJ), African-Wide Information, Global Index Medicus, and the Web of Science) for English language publications on reported NSHI causal founder variants. The search covered the prevalence, carrier frequencies, and distribution of ethnic- and population-specific founder variants implicating NSHI segregation across populations. The origin and evolutionary age of the variants traced to their most recent common ancestor were also retrieved. ETA and SMA independently conducted the literature search using selective combinations of the following keywords: “nonsyndromic, non syndromic, non-syndromic, hearing impairment, hearing loss, deafness, founder mutation, founder variant, common haplotype, origin, evolutionary history, estimated age, most common ancestor” on all databases. Additionally, the cited references in the selected articles that met the study inclusion criteria were checked and retrieved until no further study was identified. The consensus and discussions were used to resolve disagreements after the screening.

### 2.2. Selection Criteria and Data Extraction

ETA and SMA conducted the literature search exclusively from 1 July 2020 with periodic updates and only full-text English articles were considered for information retrieval. Other authors (EW and LA) reviewed the selected articles for consistency. Figure 1 shows the standard data extraction flow diagram, detailing the process of article identification, screening, eligibility, and selection. We selected 52 articles reporting 56 founder variants based on the study’s inclusion and exclusion criteria. Observational genetic studies (case–control, cohorts, cross-sectional, and case series) reporting ethnic- and population-specific founder variants associated with NSHI were selected for data synthesis. The studies that contained (1) relevant data regarding one or more associated founder variants; (2) evidence of shared haplotypes traced to a common ancestor, origin, and age; (3) estimates of the founder variant, allele and carrier frequencies, and/or prevalence; (4) enough data to compute the prevalence; (5) associated phenotype severity (HI degree and frequency threshold): (6) method used for genetic investigation and/or validation of pathogenicity; (7) heterozygous selective advantage; (8) implications for screening, diagnosis, and prognosis of the condition were critically reviewed. However, after checking for duplicates, observational studies (case reports) reporting rare variants, founder variants with associated syndromic HI phenotypes, qualitative studies, editorials, letters to editors, reviews, communications, studies with overlapping data or unavailable full-text or missing key data, in vitro studies, and animal studies, as well as non-English language publications, were excluded. Information on the evolutionary history and time of variant occurrence and their contemporary regional epidemiology is presented in this review. Geographical and population distributions of variants were further checked by querying available population- and locus-specific databases (VarSome, dbSNP, VarMap, etc). Retrieved variant descriptions, according to the Human Genome Variation Society (HGVS) nomenclature, were checked using the Mutalyzer version 2.0 software and the databases including ClinVar, ClinGen, ExAC, and gnomAD. 

### 2.3. Quality Assessment

To rule out bias, two authors (ETA and SMA), using the ascertained search criteria, performed the database search to identify publications included in this systematic review. The consistency of the selected reports was reviewed by EW and LA and any disagreement were resolved with GA and AW. The data characteristics considered for extraction and analysis include the year of publication, sample size, number of controls, founder mutation, locus, gene, coding impact, prevalence, carrier frequency, estimated age of variant, markers used for haplotype analysis, method used for age estimation, global alternate allele frequency (AAF), pattern of inheritance, ClinVar classification, age of phenotype onset, and type of defect, among many others. The data synthesis and assessment of the quality of the studies and parameters to include were performed by two independent reviewers (ETA and SMA). The Sohani et al. quality of genetic studies (Q-Genie) tool [21], and Hoy et al. risk of bias assessment tool [22] for prevalence studies were used to assess the study quality. Finally, the relevance of each study outcome informed the quality of the selected publications. 

### 2.4. Pathogenicity and Clinical Significance

Although VarSome and InterVar are both bioinformatics web-based tools, built on the American College of Medical Genetics and Genomics (ACMG)/Association of Molecular Pathology (AMP) 2015 guidelines [13], each reported founder variant classification based on the ACMG/AMP guidelines was independently reviewed. The ClinVar, dbSNP, and ClinGen classifications of variants were also examined and the allele frequencies from the ExAC and gnomAD databases were extrapolated. In addition, the phenotype-genotype evidence and the relationship between human genetic variants and phenotypes providing strong evidence for clinical significance interpretation, were equally reviewed. Decisions on the putative variants’ clinical significance were based on the pathogenic predictions obtained from the databases queried and the clinical reports.

## 3. Results

### 3.1. Publication Search Outcome

We extracted a total of 777 publications through the electronic database search. After removing duplicates and a critical screening of titles and abstracts, 121 full-text versions of relevant articles were considered for full-text review. Finally, 52 reports were considered for information retrieval, synthesis, and data analysis, as shown in the flow diagram (Figure 1). Data on over 27,959 study participants who were involved in the studies reported in 24 countries were considered for information extraction in this systematic analysis. The details of the retrieved information characteristics extracted from the selected publications in this review are presented in Table 1.

**Figure 1 genes-14-00399-f001:**
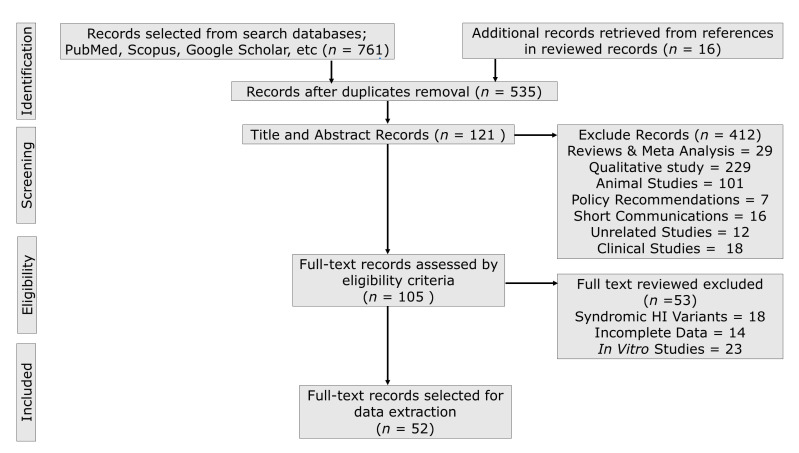
Standard information retrieval chart. The schematic flow illustrates extracted article screening and selection for data extraction, synthesis, and analysis.

### 3.2. Biased and Quality Assessment

The quality assessment scores for the articles included in the data synthesis are detailed in Appendix A. The high risk scores of possible biases for items 3, 4, and 9 (sampling method lacks randomization, non-response bias, and length of shortest prevalence period not applicable to the majority of studies reviewed (the participants are individuals with genetic conditions), respectively) were recorded for most of the publications, which is typical of most genetic studies. Of the remaining points, there was a relatively low risk of bias in the reviewed reports (Appendix A). 

### 3.3. Publications and Participants Descriptions

Since 1998, there has been an increasing trend in the number of publications that reported NSHI-associated founder variants, with a relatively high number in the past decade. This may be reflective of the current advancement in molecular diagnosis and available and cost-effective massive parallel sequencing Next Generation Sequencing (NGS) technologies. The highest number of reported NSHI-associated founder variants across populations were recorded between 2010 and 2015 (33.9%; *n* = 19/56), with considerable numbers between 2016 and 2020 as well (26.8%; *n* = 15/56) (Figure 2A). Of the studies that reported the number of probands (51.8% of studies; *n* = 29/56), there were 1341 individuals from multiplex families and 1239 simplex cases involved in the genotype and haplotype analysis for the variants investigated (2580 individuals in total). Over 76% of the reports (76.8%; *n* = 43/56) screened unrelated healthy hearing controls as comparisons for carrier frequencies of putative founder alleles, with 8426 healthy hearing controls screened in total. The age range of the few studies (28.6%; *n* = 16/56) with data on the age of the participants spanned 0–82 years (pre-lingual and post-lingual) and the mean of means for the reported ages range was 15.6 ± 4.3–42.9 ± 17.1 of the probands of 22.35 years.

### 3.4. Non-Syndromic Hearing Impairment Phenotyping

Pure tone audiometry (PTA) provides a snapshot of the level of hearing in both ears and was the most reported method used to assess the studied participants’ phenotypes in the studies reviewed. In addition, brainstem electronic response audiometry (BERA), computed tomography (CT), auditory brain stem response (ABR), and magnetic resonance imaging (MRI) were among many other assessment methods used in the phenotype screening of the participants examined. The degree of HI reported in the analyzed studies showed that the most affected individuals had mild-to-profound HI. Unilateral HI was reported in a few studies that recorded that information. While most of the affected individuals screened in the studies were reported as having non-progressive NSHI, some isolated studies (16.1%; *n* = 9/56) recorded progressive impairment in the investigated cohorts.

The pre-lingual onset of associated phenotypes was reported in 30 (53.4%; *n* = 56) studies, compared to 10 (17.9%; *n* = 56) post-lingual, and as high as 16 (28.6%; *n* = 56) reports not specifying the age of medical diagnosis in the probands investigated. Sensorineural NSHI was the most common type of defect reported (62.5%; *n* = 35/56) to segregate with pathogenic founder variants among kindreds. Although quite an appreciable number of studies (7.5%; *n* = 21/56) did not categorize the type of HI defect as either conductive or sensorineural, none of the studies reviewed attributed the NSHI (phenotype)-associated founder variant to a conductive defect in the studied probands. Virtually all the phenotypes described in the examined probands had no evidence of any other associated clinical syndromes (syndromic HI), with only one (1.9%; *n* = 56) report having no data on the phenotype classification. The autosomal recessive mode of inheritance was the most reported pattern of inheritance linked with the segregation of the reported population-specific NSHI-associated founder variants (87.5%; *n* = 49/56). Little evidence of consanguinity in some of the families investigated for generational history was reported. Autosomal dominant inheritance was also described in five (8.9%; *n* = 56) studies as associated with NSHI founder mutations and the remaining 3.6% (*n* = 2/56) of the studies did not report the specific pattern of inheritance for the reported founder variant segregation.

### 3.5. Genotyping Methods

DNA purified from peripheral blood samples was the most common source of DNA used for all the genetic investigations in the studies reviewed (96.4%; *n* = 54/56), except in two studies where buccal mucosa cells were collected in addition to the peripheral blood samples. The targeted gene sequencing approach, specifically the Sanger sequencing using sequence-specific primers, was the most widely used genotyping method in over 90% of the NSHI-associated founder mutations reported (Figure 2C). Polymerase chain reaction-restriction fragment length polymorphism (PCR-RFLP), single-strand conformational polymorphism (SSCP), whole-exome sequencing (WES), allele-specific PCR (AS-PCR), Array, real-time PCR (RT-PCR), denaturation high-performance liquid chromatography (DHPLC), and targeted panel sequencing (OtoSCOPE) were among other genotyping techniques used for the founder variant genotyping and markers for haplotype investigations in the reviewed studies.

### 3.6. Haplotype and Marker Analysis

To effectively investigate the shared haplotypes for common ancestry and the origin of the putative NSHI-associated founder variants in homozygote individuals, short tandem repeats (STRs) and single nucleotide polymorphism (SNPs) markers were genotyped to identify the shared ancestral informative markers (haplotype markers spanned between 731 bp and 63,759 bp regions around the markers of intertest). Out of the 56 reports reviewed, 20 (35.7%) estimated the evolutionary age of the reported NSHI founder variants (Appendix A). The single marker method that uses the maximum likelihood-based assumptions [71], disequilibrium mapping likelihood estimation (DMLE+v.2.3) [72], and genetic clock concept that relies on the decay of markers in linkage disequilibrium and estimates from recombination fractions were among the methods used for the age estimations in the studies reviewed. However, few studies relied on historical events in the history of the affected population to estimate the age of the founder variant. The single marker method uses the basic assumption that two affected individuals are descendants of a local ancestor who introduced the mutation some number of generations ago. Using the size of the haplotype shared by affected individuals, the unknown number of generations is estimated on each side of the locus [71]. On the other hand, the DMLE+v.2.3 population genetic software uses the Bayesian linkage disequilibrium gene mapping approach, which depends on the linkage disequilibrium between pathogenic mutation and multiple linked markers in affected individuals and unaffected controls [72]. The genetic clock principle for mutation age estimation is based on the expected decay of the linkage disequilibrium between the mutation and alleles of neighboring genetic markers due to recombination. 

### 3.7. Regional Distribution of Founder Variants

Most of the NSHI-associated founder variants were reported in Asia (85.7%; *n* = 48/56), specifically with the majority in Japan, Russia, China, South Korea, Iran, and Pakistan (Figure 2B,D and Figure 3). For instance, in Japan, seven (7) different founder mutations were reported in five (5) genes (Figure 3 and Appendix A). Europe was the continent with the second most reports (16.1%; *n* = 9/56); Africa and North America recorded 7.1% (*n* = 4/56) studies each, while South America had only one (1) publication (1.8%) (Figure 2B,D). Two NSHI-associated founder mutations were reported both in Tunisia, India, and the USA. One NSHI-associated founder mutation each was reported to originate from Ghana, Morocco, Mexico, Belgium, Guatemala, Oman, Pakistan, and Greece (Figure 3).

### 3.8. Reported Founder Mutation Genes 

The analysis of the implicated genes with reported NSHI founder variants illuminated *GJB2* as the predominant gene with the highest number of reported variants in all regions, further supporting the hypermutability hypothesis (Figure 3 and Figure 4). The contribution of other gene variants implicated in the segregation of NSHI founder alleles were region-, population-, and ethnic-specific, also suggesting the occurrence of the local founder effect and/or geographical isolation. *TMC1* and *GSDME* had two different reported variants. Of the studies reviewed, *GJB2* genetic variations remain the most extensively investigated and widely reported causal variants; with 32 different variants reported across populations, representing 57.1% of all NSHI causal variants (32/56) (Figure 2B and Appendix A). In Russia, comparative haplotype analysis favored a common origin of *GJB2*: c.35delG-p.Gly12Ter in the Volga-Ural region and Siberia and in European ancestry [34]. Subsequent investigation of other *GJB2* alleles reported at higher frequencies (c.235delC-p.Leu79CysTer3, c.516G>C-p.Trp172Cys, and c.-23+1G>A) in Southern Siberia identified a shared haplotype to further support the proffered founder role in the variant segregation in that region [39]. The *GJB2* founder variants: c.35delG (19.6%; *n* = 11/56), c.235delC (12.1%; *n* = 7/56), IVS1+1G>A (7.1%; *n* = 4/56), and c.427C>T (3.4%; *n* = 2/56) were the most common and re-occurring NSHI-associated variants with a putative founder effect across the ethnic groups. All the other reported NSHI-associated founder gene variants relate to specific populations (Appendix A).

Gasparini et al., in 2000, reported a possible common ancestor for the *GJB2*: c.35delG-p. Gly12ValfsTer2 variant and suggested it may have originated in either Europe or in the Middle East populations; the authors also proposed a probable heterozygote carrier advantage [74]. In 2001, Van Laer et al. showed that the *GJB2*: c.35delG-p.Gly12ValfsTer2 variant may have evolved centuries ago in the white population. They identified the shared haplotype around the locus in homozygote participants from Belgium, the United Kingdom, and the United States that was absent in the population-matched hearing controls negative for the variant [25]. The West Austria study by Janecke et al., in 2002, reported the highest prevalence (72.1%) of the *GJB2*: c.35delG-p.Gly12ValfsTer2 variant [26]. Some studies in Turkey identified a higher prevalence of that variant among hearing-impaired individuals compared to controls [27,75]. The *GJB2*: c.35delG-p.Gly12ValfsTer2 deletion was also reported to originate from China [28], Morocco [29] Greece [30], India [31], Iran [33,35], and Russia [32,34]. 

The *GJB2*: c.235delC-p.Leu79CysTer3 variant was also predominant among Altaians (South Siberia and Russia). Genetic drift or a founder effect was suggested as the likely underlying molecular basis for the relatively high carrier frequency (4.6%) of the variant observed [38]. However, as to whether the Altai region could be one of the multiple founder origins of the widespread *GJB2*: c.235delC-p.Leu79CysTer3 variant in Asia, was uncertain [38]. The recent report by Zytsar et al. supports the *GJB2*: c.235delC-p.Leu79CysTer3 founder variant ancestry in indigenous Southern Siberia populations, after haplotype reconstruction analysis elucidated the shared haplotype among homozygote-affected individuals [39].

The *GJB2*: c.427C>T-p.Arg143Trp missense variant was first reported as a founder mutation linked to the segregation of NSHI among a population in a village (Adamrobe) in Ghana, West Africa [11]. However, Shinagawa et al., after haplotype analysis using markers in linkage disequilibrium with the variant (*GJB2*: c.427C>T-p.Arg143Trp) in a relatively small number homozygous individuals in Japan, suggested phenotype segregation to be a likely founder source in a Japanese ancestor that evolved 6,500 years ago and not a hot spot mutation [23]. Ghanaian families segregating the *GJB2*: c.427C>T-p.Arg143Trp variant are reported to carry a distinct haplotype traced to a common indigenous ancestor some 9,625 years (385 generations) ago [76], predating the Japanese report. The deleterious allele continuance is suggested to have been sustained at the observed relatively high frequency in Ghana due to the high assortative and complementary marriages among early carriers. 

The *GJB2*: IVS1+1G>A (c.-23+1G>A) splice donor variant was also reported by publications from Siberia [39] and Turkey [61] as resulting from a founder effect traced to a common origin in the investigated NSHI families. The details of other gene variants implicated in the reported putative founder variants in different populations reported in the studies reviewed are shown in Appendix A. In South Korea, Turkey, Russia, Iran, and China, additional founder variants in different genes associated with NSHI were reported (Appendix A). 

### 3.9. Pathological Mechanisms of Implicated Genes

The systematic findings have demonstrated that congenital NSHI primarily results from developmental defects in the cochlear rather than the anticipated hair cell degeneration and endo-cochlear potential reduction [77]. The loss of function (LoF) mutation has been suggested as the main pathological mechanism reported in *GJB2* deleterious alleles (the most common cause of NSHI). Similar to this report, the molecular consequences of the reported *GJB2* founder variants, based on their impact on coding (Figure 4D) and the type of mutation (nonsense (33.9%; 19/56), missense (25.0%; 14/56), deletion (16.1%; 9/56), frameshift (16.1%; 9/56), and splice donor site (7.1%; 4/56, or a 5 prime UTR variant) supports the proposed LoF pathological mechanism (Figure 4D). Functional studies on *GJB2* variants have revealed varied pathological mechanisms as leading to HI. The proposed pathobiological consequence include: i) mutational changes that affect the correct tracking of the gap junction channels to the cell surface [78], ii) defective intercellular gap junctions [79], iii) selective permeability to ions but not small molecules [80], and iv) inability to form heterotypic channels [78], among others. The connexins are expressed chiefly in the organ of Corti’s supporting cells, stria vascularis, spiral ligaments, and limbus, as well as in other non-sensory cells and structures in the cochlear [81]. Two networks of gap junction have been identified, the epithelia gap junctional in the organ of Corti and the connective tissue gap junction network in the cochlear lateral wall. Pathological changes observed in *GJB2* knock-out mice showed cochlear development disorder, hair cell and spiral ganglion degeneration, and inactive cochlear amplification, confirming the loss of function pathology in humans [82]. 

The high prevalence of pathogenic *GJB2* alleles in families perpetuating the associated NSHI phenotype across populations regardless of ethnic group led to the suggestion that probable improved fitness (survival benefit) is conferred by these alleles. To investigate this hypothesis, Meyer et al. in 2002 reported the relative expression of *GJB2* in the skin of cohorts, with *GJB2* variants having thicker skin, which is plausibly a barrier that improves the ability to prevent pathogen invasion, trauma, and insect bites [83]. This assertion may be supported by reported *GJB2* variants on the gnomAD population database (Table 2). The anticipation is that, under natural selection, the number of ‘expected’ and ‘observed’ LoF mutations for a gene should not be significantly different (https://gnomad.broadinstitute.org/ accessed on 10 November 2022). However, this is not the case for this hypermutable *GJB2* gene, where the observed number of LoF SNVs (*n* = 17) is significantly higher than the expected number of LoF SNVs (*n* = 6.5). This observation suggests that unknown genetic modifiers (genetic constraints) and/or environmental pressure may have led to the evolution of variants in the *GJB2* gene across populations. Though haplotype analysis across populations has confirmed the independent evolution of some recurrent *GJB2* alleles and the reported allele frequency is higher than would be expected under neutrality for a pathogenic variant, to date, there is no report correlating any potentially responsible past or present defined endemic condition(s) in the history of the populations reporting deleterious founder alleles [76]. To better understand whether these predominant NSHI founder variants are under any kind of selection or coevolved to provide some fitness in early carriers, it may be helpful to further scrutinize the genome wide data of individual’s homozygous for these variants across different ethnic groups. In contrast to the distribution of *GJB2* variants on gnomAD, the number of expected LoF SNVs for the other genes were not relatively different from the observed number of LoF SNVs (Table 2).

## 4. Discussion

This study systematically reviewed the reported putative P/LP founder variants associated with NSHI worldwide. The review showed a high proportion of recessive founder variants (87.5%; *n* = 49/56) relative to dominant variants. Consistent with previous reports, Asia and Europe had the most reported NSHI founder variants, with only isolated reports in North America and Africa [36,84]. This observation may reflect disparities in research priorities and funding distribution and could partly be linked to the epidemiology of HI burden in Asia and the Middle East, as well as the prevalent cultural consanguinity practices in the region [17,28].

### 4.1. GJB2 Founder Variants

The finding equally highlights *GJB2* as the most studied and recurrent NSHI gene, implicated with over 57.1% NSHI founder variants, which further demonstrates the gene’s variants as the most predominant cause of NSHI worldwide [85]. This observation intensifies the need to generate a haplotype map on the populations reporting specific NSHI founder variants of multiple origins to clarify the timing and likely evolutionary constraints (modifiers) influencing the variants’ segregation across populations.

*GJB2* hypermutability [86,87,88,89] and the proposed heterozygote selective carrier advantage [83] has been linked to improved fitness for the continuous perpetuation of the phenotype across ethnic groups. Interestingly, the probability of the *GJB2* LoF score, observed / expected (o/e) on gnomAD (https://gnomad.broadinstitute.org/ accessed on 10 November 2022) favors the survival benefit hypothesis proffered on deleterious founder alleles [83]. The pLoF score is a continuous measure that describes a gene’s tolerance to certain classes of genetic variation. The high *GJB2* (o/e) pLoF score of 2.62 (90% CI: 1.39–1.98) on gnomAD shows relatively higher observed LoF variants recorded over the years relative to expected LoF variants, suggesting that *GJB2* may be under selection for LoF variants. However, since the pLoF value of 2.63 is outside the 90% credible interval (90% CI: 1.39–1.98), the gene is distinguished as one of the peculiar cases where there are a lot of uncertainties about the constraint metrics due to the number of samples and gene size. On the other hand, a lower o/e score may have suggested a stronger selection against the class of LoF variants, relative to a higher o/e score. However, since the variant count is dependent on the gene size and sample size, relying on the precision of the o/e score may be limited for absolute comparison with other genes. For example, considering the size of *GJB2* (226 aa) against the huge number of variants reported (64 synonymous and 161 non-synonymous variants), further investigations with enough power may be required to elucidate the genetic constraints at the locus of this gene. 

Haplotype analysis investigating shared genetic ancestry demonstrated independent multiple origins for some founder variants across ethnic groups. Inheriting the homozygous or compound heterozygous state of the *GJB2*: c.35delG-p.Gly12ValfsTer2 variant accounted for 10–20% of all congenital HI in American Caucasians from North and South Europe [26,27,75]. Haplotype analysis of the SNPs in linkage disequilibrium with the *GJB2*: c.35delG-p.Gly12ValfsTer2 variant traced the variant’s origin to the Middle East, before spreading to Europe and subsequently throughout the region [25]. Further analysis that computed the haplotypes in homozygote cohorts in Turkey and Iran also suggested a single origin in Anatolia and Iran [27]. In China, improved reproductive fitness and ethnic group inter-marriage was cited among the underlying genetic forces for the observed high (11.5%) carrier frequency of the variant [88] reported in Chinese cohorts. Though contentious, the variant may have originated in European and North American ancestors or Middle East populations about 100 centuries ago, before spreading to other regions [25]. Alternatively, perhaps the assertion that it first arose in Central Asia before it was transferred roughly 11,800 years ago [60] needs more attention?

Investigating the *GJB2*:c.35delG-p.Gly12ValfsTer2 variant among Moroccan cohorts also presented a shred of strong evidence of three shared markers in linkage disequilibrium with the variant, suggesting a possible evolutionary ancestor in Morocco [29]. This report favors the idea of multiple independent origins of the variant as opposed to an earlier report that the *GJB2*: c.35delG-p.Gly12ValfsTer2 variant originated from a single Mediterranean ancestor some 10,000 years or nearly 500 generations ago [25]. Ascertaining global haplotypic backgrounds in mutant chromosomes will help identify an integral panel of haplotype diversity and backgrounds sustaining the variant segregation to disambiguate the route of spread of the *GJB2*: c.35delG-p.Gly12ValfsTer2 variant. Generating data on the mutational load across populations reporting the variant using NGS data may illuminate the haplotypic backgrounds carrying the variant and elucidate potential modifiers of phenotype variability. This will, in addition, identify the likely shared genetic ancestry and common backgrounds sustaining the variant distribution across populations.

The *GJB2*: c.235delC-p.Leu79CysfsTer3 variant was the second most common and recurrent *GJB2* reported founder variant. The variant accounted for 65.5% of all *GJB2* mutant alleles and is considered the most prevalent NSHI-associated variant in Asia [90]. The markers in high-linkage disequilibrium with *GJB2*: c.235delC-p.Leu79CysfsTer3 demonstrated in haplotype screening, traced the variant’s origin to a common founder in Altaians, Mongolians, Chinese, and Japanese [25,73]. In Mongolia, the concurrence of the *GJB2*: c.35delG-p.Gly12ValfsTer2, c.235delC-p.Leu79CysfsTer3, and c.-23+1G>A variants in the study participants [36] reflected the geographical setting of the population as a crossroad in the migration history. Yan et al. estimated the variant to have evolved in the Baikal area some 11,500 years before it spread to Mongolia, China, Korea, and Japan by human migration [36]. Although this estimate mirrors the Van Laer et al. report on the c.35delG variant [25], the Altai region could be the possible origin of the *GJB2*: c.235delC-p.Leu79CysfsTer3 founder variant, before expanding across Asia [36]. The observed discrepancies in the origin and estimated age of the *GJB2*: c.235delC-p.Leu79CysfsTer3 variant in different populations could result from the differences in the age estimation methods, the underlying principles and assumptions, specific ancestral informative markers used for the haplotype analysis, sample size differences, population characteristics, and distinct bottle-neck events in the populations’ history and demographic events [39]. 

The *GJB2*: c.-23+1G>A (IVS1+1G>A) variant accounted for 95% of all deleterious *GJB2* variants examined in the Yakut-affected cohorts [60]. A reconstruction of 140 haplotypes with the *GJB2*: c.-23+1G>A (IVS1+1G>A) variant suggested a common ancestor in all the homozygote chromosomes, further supporting the founder source. The estimated age of the variant (~800 years) is consistent with the Turkic-speaking Yakut ancestor’s history in Eastern Siberia [91]. Computing the haplotype in homozygous Turks and Mongolians predicted Central Asia as the origin of the variant before migration spread the carriers of the variant to the Middle East [27]. Considering the Tuvinians’ history of Mongolian influence at different periods in their ethnic formation [92], the variant may have been introduced into Tuva by ancient Mongolians about 725–4100 years ago, before moving to Siberia. The c.-23+1G>A variant has been reported at varied frequencies (from 0.6% to 58.1%) across populations, from East Asia to Russia and Central Asia, however, there are no reports of the variant in Southeast Asia and sub-Saharan Africa.

The *GJB2*: c.167delT-p.Leu56ArgfsTer26 variant is the most common pathogenic *GJB2* variant in the Ashkenazi Jewish population [36,93]. The hypothesis for a shared origin of this variant within deaf Palestinians and Israelis was supported by the identification of significant markers that were in strong linkage disequilibrium with the *GJB2*: c.167delT-p.Leu56ArgfsTer26 variant in the investigated cohorts [93]. Morell et al. observed the previously reported highly conserved haplotype markers in the linkage disequilibrium with the variant and proposed a single evolutionary origin among Ashkenazi Jews [36]. 

The *GJB2*: c.427C>T-p.Arg143Trp variant remains the most familiar *GJB2* variant associated with NSHI segregation in Ghana, with as high as 37% prevalence [94]. Isolated reports of the variant, though at relatively low frequencies, in East Asia, Europe, and the Middle East traced the origin to Japanese ancestry from about 6,500 years ago [23,95]. This suggests selection at this locus in other populations, together with a predominant founder effect reported in Ghana [10,11,12]. The computed haplotype analysis that compared the *GJB2*: c.427C>T-p.Arg143Trp homozygous Ghanaians and SNV data extracted from the 1000 Genomes populations revealed that *GJB2*:c.427C>T-p.Arg143Trp is carried on different haplotype backgrounds in Ghanaian and other populations, particularly in Japan, as well as among populations of European ancestry, providing further support to the multiple independent origins hypothesis [76]. Remarkably, the recent identification of *GJB2*: c.427C>T-p.Arg143Trp in 4.5% (*n* = 2/44 families) among a Senegalese cohort further strengthens the need to explore more populations for *GJB2* pathogenic variants in West Africa [96]. This report raises questions on the ancestral and migration histories between Ghana and Senegal. Though the variant has been extensively investigated in Nigerian [97], Cameroonian [98], and Malian [99] HI cohorts, the variant had no contribution to the phenotype in these populations. Considering the distance between Ghana and Senegal relative to the proximity of Nigeria, Cameroon, and from Mali to Ghana, this report suggests that transatlantic slave transport along the coastal regions of West Africa, drained by forts in Senegal, is likely to have led to the variant’s spread to Senegal. However, investigating the haplotypic backgrounds in homozygous Ghanaian and Senegalese cohorts will clarify the hypothesized route of the variant’s spread.

The observed deficit of reports on P/LP NSHI variants’ prevalence in Africa/African Americans relative to other populations in gnomAD, evident in this report, may reflect the limited HI genetic studies and lack of follow-up on clinical cases for genetic testing in West Africa. 

### 4.2. Other Genes (Non-GJB2) Founder Variants

The haplotype analysis of the SNPs in linkage disequilibrium with the transmembrane inner ear gene (*TMIE*) variant, *TMIE*: c.250C>T-p.Arg84Trp, demonstrated a single origin traced to about 1250 years ago in Turkey. Non-complementary deaf-by-deaf marriage, prevalent in families in Southeastern Anatolia, was linked with this variant segregation in all the affected siblings. The proposed ancestor is thought to have lived after the Arabs came to Southeastern Anatolia and migration possibly introduced the variant into the population [100]. The 3-base polypyrimidine deletion in intron 7 of the gasdermin E gene (*GSDME*), *GSDME*: c.991-15_991-13delTTC, was reported as a founder variant in East Asian families in Korea, China, and Japan [65,66]. Park et al. demonstrated a putative founder effect for the variant after comparing the variant-linked haplotypes between Chinese and Korean families [65]. However, this report does not support the European ancestry hypothesis, suggesting a possible hotspot for the mutation due to records of the variant in Toscani, Ashkenazi Jews, and Hungarian families [67]. Haplotype investigation for a common ancestor of the transmembrane serine protease 3 gene (*TMPRSS3*) variant, *TMPRSS3*: c.916G>A-p.Ala306Thr, established markers in linkage disequilibrium, suggesting an ancestral founder in China [53]. The phenotype associated with the *TMPRSS3* founder variant was consistent with a previous report of inter-familial and intra-familial variability, showing a progressive phenotype [101]. An initial report had described the potassium voltage-gated channel subfamily Q member 4 gene (*KCNQ4*) variant, *KCNQ4*: c.211delC, as stemming from a possible hot spot, favorable to mutations rather than the proposed common founder in Japan [63]. Haploinsufficiency and dominant-negative effects were hypothesized as possible mechanisms in the pathophysiology and associated phenotype variability. However, the elucidation of this rare mutation in two Lak families, separated by a geographical barrier, led to the founder effect hypothesis [63]. Carriers of the pejvakin gene (*PJVK or DFNB59*) variant, *PJVK*: c.406C>T-p.Arg136Ter, were reported in two extended families whose ancestors migrated from Sudan to Israel over 150 years ago [3]. The incidence of intellectual disability recorded in the families restricted other families from marrying into these two families [102], segregating the founder variant. The high prevalence (56.5%; *n* = 13/23) of the otoferlin gene (*OTOF*) variant, *OTOF*: c.5816G>A-p.Arg1939Gln, led to the founder source being presumed as having evolved in a single Japanese ancestor [54]. The haplotype analysis found that all the affected chromosomes homozygous for the PDZ domain containing the 7 gene (*PDZD7*) variant, *PDZD7*: c.490C>T-p.Arg164Trp, shared six (6) STRs in linkage disequilibrium. This, together with the observed high carrier frequency, demonstrates a possible founder source relative to the proposed hot spot prone to mutations for the NSHI-associated phenotype segregation [59]. The high level of inbreeding in Oman, coupled with the demographic explosion in the last decade, may have contributed to the myosin XVA gene (*MYO15A*) variant, *MYO15A*: c.1171_1177dupGCCATCT-p.Tyr393CysfsTer41, duplication and subsequent diffusion in the population [46]. The authors hypothesized East Asia is the likely origin and founder source of the variant *CDH23*: c.C719T-p.Pro240Leu based on the high frequency of the allele in the Korean pediatric population 85.7% (6/8) [55]. This assertion was confirmed in the haplotype analysis of the *CDH23*: c.C719T-p.Pro240Leu variant that found the variant to be carried on a single haplotype linked to the variant, completely absent in normal-hearing controls [55]. The analysis of the differences between the *CDH23*: c.C719T-p.Pro240Leu variant frequencies of East Asia and other groups showed the variant arose from Central to East Asia about 40,000 years ago [103]. The haplotype analysis suggested that the claudin 14 gene (*CLDN14*) variants, *CLDN14*: c.254T>A-p.Val85Asp and c.398T>G-p.Met133ArgfsTer23, as founder variants originated from Pakistan [56]. The observed inter- and intra-familial phenotype variability among the affected kindreds with the same *CLDN14* variants were linked to environmental factors and genetic modifiers [56]. The recent Korean study that analyzed *CDH23*: c.857delC-p.Asn2858GluTer8 and linked microsatellite markers reported a common founder for the variant and suggested evidence of allele decay in the haplotype analysis [55]. The authors argued that, if the suggested founder variant had evolved some time ago, genetic recombination may likely explain the observed shortening in a proportion of the analyzed haplotypes.

### 4.3. General Observations on Founder Variants

Generally, the observed population- and ethnic-specificity of most of the reported NSHI-associated gene variants suggest a possible underlying heterozygous carrier advantage that may be protective against indigenous infection, confer disease protection, and/or provide nutritional benefits [83]. Irrespective of the impact of a variant on fitness, most disease-associated founder variants occur at relatively high frequencies, culminating from the effects of possible random genetic drift—likely after a historical bottleneck event—that is subsequently fueled by consanguineous unions leading to the observed incidence of monogenic recessive disorders in some populations [104]. This phenomenon could be extended to observations in Asian and Middle Eastern populations with high consanguinity rates. Dynamics such as the size of the founding population, level of isolation due to geographical location, ethnicity, and existing cultural barriers may have contributed to the epidemiology of NSHI-associated founder variant segregation across populations [105]. Moreover, admixture and sporadic mutations may have caused further segregation of some recessive alleles, specifically for *GJB2* founder variants [106]. 

However, data that explore the precise mechanism (proposed combination of improved genetic fitness, assortative mating, and unknown genetic modifiers) for the selection or selective sweep of reported HI deleterious alleles among families, and the seemingly heterozygote carrier advantage, remains to be generated [107,108].

## 5. Conclusions

This report further emphasizes pathogenic variants in *GJB2* as the most prevalent cause of NSHI, widely reported at characteristic high frequencies in Asia, Europe, and North Africa, with sub-Saharan Ghana as the African exception. The study highlights the dearth of HI genetic investigation in Africa, particularly on haplotype conservation in families with suspected founder variants. Considering the observed discrepancies in the origin and estimated age of some reported NSHI-associated founder variants, future studies should focus on the computation of common ancestry haplotypes in the homozygous mutant chromosomes of the predominant founder variants, using Next Generation Sequencing data. We encourage efforts to explore putative founder variants to ascertain allele frequencies and regional distributions and, thus, to facilitate policy recommendations for genetic counselling and genetic testing. 

## Figures and Tables

**Figure 2 genes-14-00399-f002:**
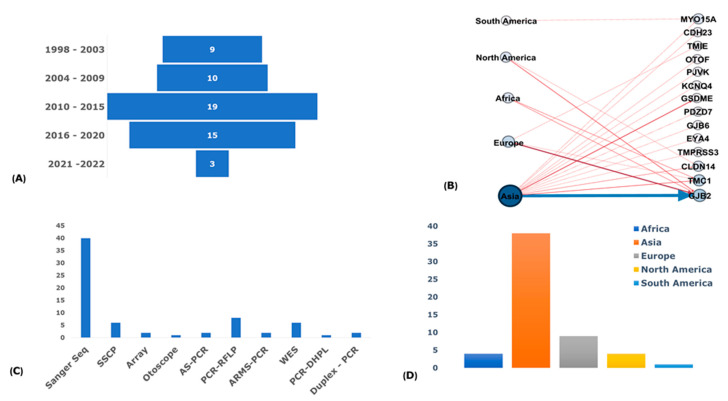
Distribution profile of NSHI-associated genes with founder variants; (**A**) A bar chart showing the distribution of publications retrieved per year range reporting NSHI-associated P/LP founder variants between 1998 and 2022. (**B**) A network of reported genes with the mutations plotted against continents from which the mutations were reported. The continents denote the nodes on the left and nodes on the right represent the individual genes harboring the PLP reported founder mutations. The thick blue line depicts *GJB2* as the gene with most reported mutations across the continents. The network was constructed using the Gephi software. (**C**) Genotyping methods were used to investigate founder mutations for common ancestor or haplotype sharing in individuals segregating the ethnic- and population-specific PLP NSHI founder mutations. At least one of the methods listed was used for the genotype analysis in each of the studies reviewed: Sanger Seq = Sanger sequencing, SSCP = Single-strand conformational polymorphism analysis, Array = Affymetrix GeneChip Array, Otoscope = OtoSCOPE platform, AS-PCR = Allele-specific polymerase chain reaction, PCR-RFLP = Polymerase chain reaction-restriction fragment length polymorphisms, ARMS-PCR = Amplification-refractory mutation system, WES = Whole exome sequencing, PCR-DHPLC = Polymerase chain rection Denaturing High Performance Liquid Chromatography, and Duplex, and PCR = Duplex polymerase chain reaction. (**D**) A bar chart of the distribution of reviewed articles by continents.

**Figure 3 genes-14-00399-f003:**
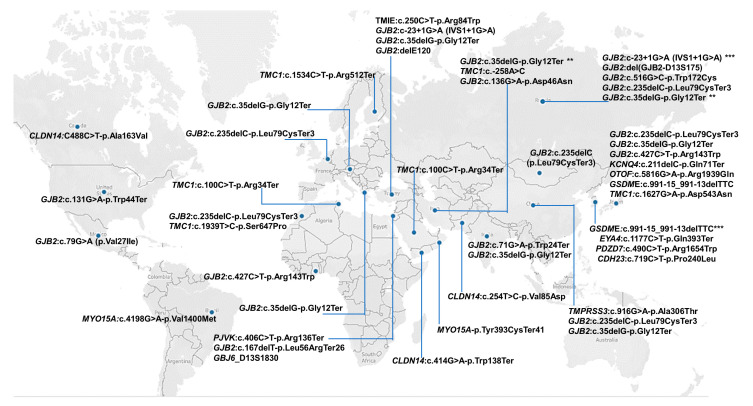
A global map showing countries that reported at least one gene with P/LP founder variant in NSHI. Ethnic- and country-specific reported NSHI founder mutations are depicted with blue dots, with the reported genes and mutations listed in specific countries, respectively. The map was created using Tableau Software [73]. * = number of times variant has been reported as founder.

**Figure 4 genes-14-00399-f004:**
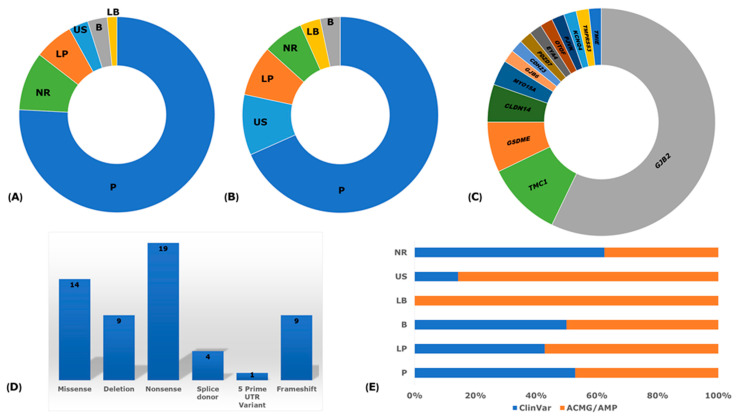
Reported variant classification. (**A**) ClinVar classification of the reported founder variants. (**B**) American College of Medical Genetics and Genomics (ACMG) and Association of Medical Pathology (AMP) guideline classification of reported founder variants. (**C**) A pie chart showing the different genes and proportions reported for NSHI founder mutations. (**D**) The number, proportions, and types of variants reported. (**E**) ClinVar and ACMG/AMP proportions compared. P = Pathogenic; LP = Likely pathogenic; B = Benign; LB = Likely benign; US = Uncertain significance; and NR = Not reported. Note: (**A**) and (**B**) were normalized by adding “+1” to all values obtained for each category to obtained consistent color coding of sections.

**Table 1 genes-14-00399-t001:** NSHI founder mutations allele frequency global distribution, ClinVar, and ACMG/AMP classifications.

							ExAC Freq		GnomAD Exomes Freq		
Reference SNP	Transcript	Locus	Founder Variant	Coding Impact	ACMG/AMP	ClinVar	Ref. Allele	Alt. Allele	Ref. Allele	Alt.Allele	Ref
rs80338948	NM_004004.6	DFNB1	*GJB2*: c.427C>T-p.Arg143Trp	Missense	P (PP5, PM1, PM2, PP3, PM2)	P	0.999835	0.000165	0.999884	0.000116	[11,23]
rs80338942	NM_004004.6	DFNB1	*GJB2*: c.167delT-p.Leu56ArgTer26	Frameshift	P (PSV1, PP5)	P	0.999316	0.000684	0.999415	0.000585	[24]
rs80338939	NM_004004.6	DFNB1	*GJB2*: c.35delG-p.Gy12Ter	Nonsense	P (PSV1, PM2, PP5)	P	0.993960	0.006040	0.993443	0.006557	[25,26,27,28,29,30,31,32,33,34,35]
rs80338943	NM_004004.6	DFNB1	*GJB2*: c.235delC-p.Leu79CysTer3	Frameshift	P (PSVB1 PS3, PP5, PM2)	P	0.999637	0.000363	0.999531	0.000469	[25,28,36,37,38,39]
NA	NA	DFNB1	*GJB6*: ∆(GJB6-D13S1830)	Deletion	NR	NR	NR	NR	NR	NR	[40,41]
NA	NA	DFNB1	*GJB2*: delE120	Deletion	NR	NR	NR	NR	NR	NR	[42]
rs104894396	NM_004004.6	DFNB1	*GJB2*: c.71G>A-p.Trp24Ter	Nonsense	P (PSV1, PP55, PM2)	P	0.999423	0.000577	0.999416	0.000584	[43]
rs2274084	NM_004004.6	DFNB1	*GJB2*: c.79G>A-p.Val27Ile	Missense	B (BA1, PP6, BP4, PM1)	B	0.954619	0.045381	0.945994	0.054006	[44]
rs104894413	NM_004004.6	DFNB1	*GJB2*: c.131G>A-p.Trp44Ter	Nonsense	P (PS3, PM1, PP3, PM5, PM2)	P	0.999992	0.000008	0.999976	0.000024	[45]
NA	NA	DFNB1	*GJB2*: del(GJB2-D13S175)	Deletion	NR	P	NR	NR	NR	NR	[1]
rs1302739538	NM_004004.6	DFNB1	*GJB2*: c.516G>C-p.Trp172Cys	Missense	P (PP5, PM1, PM2, PP3, PM2)	P/LP	NMR	NR	NR	NR	[39]
rs1567620939	NM_016239.4	DFNB3	*MYO15A*: c.1171_1177dupGCCATCT -p.Tyr393CysTer41	Frameshift	P (PVS1, PP5, PM2)	P	NR	NR	NR	NR	[46]
rs749136456	NM_016239.4	DFNB3	*MYO15A*: c.4198G>A-p.Val1400Met	Missense	P (PP5, PP3, PM2,BP1)	LP	NR	NR	NR	NR	[47]
rs28942097	NM_147196.3	DFNB6	*TMIE*: c.250C>T-p.Arg84Trp	Missense	P/LP (PP5, PM3, PP3, PM2)	P/LP	0.999983	0.000017	0.999976	0.000024	[48]
rs937270834	NM_138691.3	DFNB7/11	*TMC1*: c.-258A>C	5 Prime UTR Variant	LB (BP4, PM2)	NR	NR	NR	NR	NR	[49]
rs121908073	NM_138691.3	DFNB7/11	*TMC1*: c.100C > T-p.Arg34Ter	Nonsense	P (PVS1, PP5, PM2)	P	0.999948	0.000052	0.999944	0.000056	[50,51,52]
rs181949335	NM_032404.2	DFNB8	*TMPRSS3*: c.916G>A-p.Ala306Thr	Missense	P (PP5, PM1, PM5, PP3)	P/LP	0.999827	0.000173	0.999855	0.000145	[53]
rs80356605	NM_194248.3	DFNB9	*OTOF*: c.5816G>A-p.Arg1939Gln	Missense	US (PP5, PM2, PP3, BP1)	P	NR	NR	0.999957	0.000043	[54]
rs121908354	NM_022124.6	DFNB12	*CDH23*: c.C719T-p.Pro240Leu	Missense	LP (PP5, PM1, PM2, BP4)	P	0.999909	0.000091	0.999960	0.000040	[55]
rs74315437	NM_012130.4	DFNB29	*CLDN14*: c.254T>A-p.Val85Asp	Missense	US (PM5, PM2, PP3)	P	0.999992	0.000008	1.000000	0.000000	[56]
rs143797113	NM_012130.4	DFNB29	*CLDN14*: c.488C>T-p.Ala163Val	Missense	US (PM2, PP5)	US, LP	0.999744	0.000256	0.999708	0.000292	[57]
rs142846225	NM_012130.4	DFNB29	*CLDN14*: c.414G>A-p.Trp138Ter	Nonsense	P (PVS1, PP5, PM2)	NR	0.999992	0.000008	0.999988	0.000012	[58]
rs200664140	NM_001195263.2	DFNB57	*PDZD7*: c.490C > T-p.Arg164Trp	Missense	US (PM2, PP5, BP1)	US	NR	NR	NR	NR	[59]
rs367688416	NM_001042702.5	DFNB59	*PJVK*: c.406C>T-p.Arg136Ter	Nonsense	P (PSV1, PP5, PM2)	P	0.999982	0.000018	0.999984	0.000016	[3]
rs80338940	NM_004004.6	DFNBA1	*GJB2*: c.-23+1G>A	Splice donor	P (PSV1, PP5. PM2)	P	NR	NR	0.999829	0.000171	[39,60,61,62]
rs80358272	NM_004700.4	DFNA2	*KCNQ4*: c.211delC-p.Gln71SerTer68	Nonsense	P (PSV1, PM2, PP5)	P	NR	NR	NR	NR	[63]
rs1064797088	NM_004004.6	DFNA3A	*GJB2*: c.136G>A-p.Asp46Asn	Missense	P (PM1, PM5, PP3, PP5, PM2)	P	NR	NR	NR	NR	[50]
rs727505273	NM_001127453.2	DFNA5	*GSDME*: c.991-15_991-13delTTC	Deletion	LP (PP5, PM2, BP4)	P	NR	NR	NR	NR	[64,65,66,67]
rs757172581	NM_004100.5	DFNA10	*EYA4*: c.1177C>T-p.Gln393Ter	Nonsense	P (PVS1, PM2, PP5)	NR	0.999992	0.000008	0.999996	0.000004	[55]
rs138527651	NM_138691.3	DFNA36	*TMC1*: c.1939T>C-p.Ser647Pro	Missense	US (PP5, PM2, PP3)	P	0.999992	0.000008	0.999976	0.000024	[68]
rs200171616	NM_138691.3	DFNA36	*TMC1*: c.1534C>T-p.Arg512Ter	Nonsense	P (PVS1, PM2, PP3, PP5)	P	0.999802	0.000198	0.999670	0.000330	[69]
NA	NM_138691.3	DFNA36	*TMC1*: c.1627G>A-p.Asp543Asn	Missense	US (PM2, PP3)	US, LP	NR	NR	NR	NR	[70]

dbSNP = Single Nucleotide Polymorphism Database, ExAC = Exome Aggregation Consortium, gnomAD = Genome Aggregation Database, ClinVar = Public archive of genomic variation as it relates to human health, ACMG/AMP = American College of Medical Genetics and Genomics and Association of Medical Pathology, Ref. Allele = Reference Allele, Allele Freq. = Allele frequency, NA = Not Available; P = Pathogenic; LP = Likely pathogenic; B = Benign; LB = Likely benign; US = Uncertain significance; and NR = Not reported.

**Table 2 genes-14-00399-t002:** Founder variant(s) gene constraint metrics extracted from genome aggregation database.

Gene	Category	Expected SNVs	Observed SNVs	Constraint Metrics
*GJB2*	Synonymous Missense pLoF	61.1 137.1 6.5	64 161 17	Z = −0.29 o/e = 1.05 (0.86–1.29) Z = −0.72 o/e = 1.17 (1.03–1.34) pLI = 0 o/e = 2.62 (1.39–1.98)
*GJB6*	Synonymous Missense pLoF	64 149.6 9.3	69 134 10	Z = 0.99, o/e = 1.08 (0.89–1.32) Z = 0.45, o/e = 0.9 (0.78–1.03) pLI = 0, o/e = 1.07 (0.66–1.74)
*GSDME*	Synonymous Missense pLoF	116.1 265.4 21.6	128 304 16	Z = −0.87, o/e = 1.1 (0.95–1.28) Z = −0.84, o/e = 1.15 (1.04–1.26) pLI = 0, o/e = 0.74 (0.5–1.13)
*CDH23*	Synonymous Missense pLoF	301.3 715.2 47.1	290 662 18	Z = 0.51, o/e = 0.96 (0.87–1.06) Z = 0.71, o/e = 0.93 (0.87–0.99) pLI = 0, o/e = 0.38 (0.26–0.57)
*CLDN14*	Synonymous Missense pLoF	77 154.9 6.2	81 148 6	Z = −0.35, o/e = 1.05 (0.88–1.26) Z = 0.2, o/e = 0.96 (0.83–1.09) pLI = 0, o/e = 0.97 (0.52–1.75)
*MYO15A*	Synonymous Missense pLoF	886.4 2057.4 169.1	826 1975 116	Z = 1.6, o/e = 0.93 (0.88–0.99) Z = 0.65, o/e = 0.96 (0.92–1) pLI = 0, o/e = 0.69 (0.59–0.8)
*PDZD7*	Synonymous Missense pLoF	138.9 330 22.5	134 139 17	Z = 0.32, o/e = 0.96 (0.84–1.11) Z = −0.37, o/e = 1.06 (0.97–1.16) pLI = 0, o/e = 0.76 (0.52–1.13)
*EYA4*	Synonymous Missense pLoF	122.5 333.4 7.7	276 103 10	Z = 1.38, o/e = 0.84 (0.72–0.99) Z = 1.12, o/e = 0.83 (0.75–0.92) pLI = 0.05, o/e = 0.27 (0.16–0.45)
*PJVK*	Synonymous Missense pLoF	64.9 186.1 19.1	57 181 14	Z = 0.77, o/e = 0.88 (0.71–1.09) Z = 0.13, o/e = 0.97 (0.86–1.1) pLI = 0, o/e = 0.73 (0.48–1.15)
*TMC1*	Synonymous Missense pLoF	142.1 398.2 47.6	146 350 35	Z = −0.26;, o/e = 1.03 (0.9–1.18) Z = 0.86, o/e = 0.88 (0.81–0.96) pLI = 0, o/e = 0.74 (0.56–0.97)
*KCNQ4*	Synonymous Missense pLoF	179 412.2 32.1	156 288 7	Z = 1.35, o/e = 0.87 (0.76–0.99) Z = 2.17, o/e = 0.7 (0.63–0.77) pLI = 0.47, o/e = 0.22 (0.12–0.41)
*TMPRSS3*	Synonymous Missense pLoF	101.9 253.3 24.3	94 212 17	Z = 0.62, o/e = 0.92 (0.78–1.09) Z = 0.92, o/e = 0.84 (0.75–0.94) pLI = 0, o/e = 0.7 (0.48–1.05)
*TMIE*	Synonymous Missense pLoF	31.2 75.3 5.7	25 74 6	Z = 0.87, o/e = 0.8 (0.58–1.12) Z = 0.05, o/e = 0.98 (0.81–1.19) pLI = 0, o/e = 1.05 (0.57–1.81)
*OTOF*	Synonymous Missense pLoF	508.8 1223.6 105.2	550 1252 76	Z = −1.44, o/e = 1.08 (1.01–1.16) Z = −0.29, o/e = 1.02 (0.98–1.07) pLI = 0, o/e = 0.72 (0.6–0.88)

pLoF = Probability of loss of function; pLI = Probability of loss of intolerance; o/e = observed/expected; and Z = z – score; SNVs = single nucleotide variants

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
