# Peer review of "Global Distribution of Founder Variants Associated with Non-Syndromic Hearing Impairment"

_genes, 2023, doi:10.3390/genes14020399_

Round 1

Reviewer 1 Report

An article by Elvis Twumasi Aboagye et al. describes the genetic etiology of non-syndromic hearing loss (NSHI). In total, more than 124 different genes have been identified, which greatly complicates the molecular genetic diagnosis of this pathology in families.

The authors systematized the literature data on all identified gene variants causing NSHI, their global distribution, and the origin of the founder variants associated with NSHI.

Fifty-two (52) reports were analyzed involving 27,959 study participants from 24 countries reporting fifty-six (56) major pathogenic or probable pathogenic (P/LP) variants in 14 genes. As a result, it was found that the largest number of founder variants of NSHI was found in Asia, 85.7% (48/56) with variants in all 14 genes, followed by Europe - 16.1% (9/56) and the lack of information about variants in Africa.

The review is comprehensive and summarizes all known literature data on the genetic etiology of non-syndromic hearing loss (NSHI).

Author Response

Many thanks for your rapid and positive review.

Reviewer 2 Report

The authors have have reviewed the distribution and provided a global profile of reported NSHI founder variants prevalence, origin, and estimated age across populations. Sufficient background has been provided in the introduction. GJB2 has been well studied but this paper adds more information regarding other genes that sometimes are sidelined. 

The methods have followed appropriate guidelines and is reproducible. The methods have been well described with high quality journals being analysed for the meta-data analyses. It is difficult to interpret some of the data due to size of the font/colours/volume. Some of the conclusions are obvious based on what we already know in the reported literature already. 

It would be interesting to know what founder variants are being used in current routine screening tests and which one are currently excluded (if this data exists - NCBI gene tests). 

Figure 3  - great image. This clearly shows the data that you have generated.  

Minor corrections:

Line 25 – gene names should be in italics.

Why were the 14 genes out of 124 selected for analysis?

Include which genes are dominant vs recessive NSHI. Also, it would be helpful to have alignment with their appropriate DFNA and DFNB code as many scientists may only work on one NSHI gene and not be familar with the other listed genes. I note that this is in the supplementary but would be useful in the main text of the review somewhere.

Table 1: 

(1)   include headers on each page. 

(2)   Its unclear how this table has been arranged so rs28942097 is reported first. Thus, I’m overwhelmed on how to make any sense of the data in terms of themes.

Figure 2 

(A) Make y-axis label font bigger

(B) I don’t see a thick green line depicting GJB2 as the gene with most reported mutations across the continents. Need to correct.

(C & D) Make the x-axis label font bigger.

Figure 4

Make (C) pie chart font labels bigger.

Author Response

Comments

Response/Changes/Revision

Line 25 – gene names should be in italics.

Corrected, gene symbols in italics.

Why were the 14 genes out of 124 selected for analysis?

The 124 genes were in reference to all the identified genes implicated in non-syndromic hearing impairment in literature, and the 14 genes discussed are the reported genes with founder variants from this review.

Include which genes are dominant vs recessive NSHI. Also, it would be helpful to have alignment with their appropriate DFNA and DFNB code as many scientists may only work on one NSHI gene and not be familar with the other listed genes. I note that this is in the supplementary but would be useful in the main text of the review somewhere

Dominant and recessive pattern of inheritance proportions are now provided in the results, and respective genes indicated in Table S1.

DFNA and DFNB classification provided in column three (3) of Table 1.

(1) Include headers on each page

(2) Its unclear how this table has been arranged so rs28942097 is reported first. Thus, I’m overwhelmed on how to make any sense of the data in terms of themes

Table headers inserted as suggested.

Variants in Table 1 are in increasing order according of the year they were first published,

Variants in Table S1 are also reported in the same way.

Figure 2 (A) Make y-axis label font bigger (B) I don’t see a thick green line depicting GJB2 as the gene with most reported mutations across the continents. Need to correct. (C & D) Make the x-axis label font bigger.

Figure 2 (A) Font Size are now increased.

Figure 2 (B): Green is now changed to Blue.

Figure 2 (C&D): the font is now font increased.

Figure 4

Make (C) pie chart font labels bigger.

Figure 4 (C) Pie Chart font labels are increased.

On this comment of interest: It would be interesting to know what founder variants are being used in current routine screening tests and which one are currently excluded (if this data exists - NCBI gene tests).

Since non-syndromic hearing impairment (NSHI) is genetically heterogenous, with a wide spectrum of implicated genes, available genetic testing utilizes gene panels, array, or gene chip containing variants in known HI gene and or whole exome sequencing preferably, generally after genotyping for predominant population specific founder variants; Europe, America, Asia, and Northern Africa screen for the GJB2: 35delG-p.(Gly12Ter2) before any further testing.
